# Clinical Differentiation between a Normal Anus, Anterior Anus, Congenital Anal Stenosis, and Perineal Fistula: Definitions and Consequences—The ARM-Net Consortium Consensus

**DOI:** 10.3390/children9060831

**Published:** 2022-06-03

**Authors:** Eva E. Amerstorfer, Eberhard Schmiedeke, Inbal Samuk, Cornelius E. J. Sloots, Iris A. L. M. van Rooij, Ekkehart Jenetzky, Paola Midrio

**Affiliations:** 1Department for Pediatric and Adolescent Surgery, Medical University of Graz, 8036 Graz, Austria; eva.amerstorfer@medunigraz.at; 2Clinic for Paediatric Surgery and Paediatric Urology, Klinikum Bremen Mitte, 28205 Bremen, Germany; Eberhard.schmiedeke@klinikum-bremen-mitte.de; 3Department of Pediatric and Adolescent Surgery, Schneider Children’s Medical Center of Israel, 4920235 Petach Tikva, Israel; inbal.samuk@gmail.com; 4Sackler Faculty of Medicine, Tel Aviv University, 6997801 Tel Aviv, Israel; 5Pediatric Surgery Department, Erasmus MC-Sophia Children’s Hospital, 3015 CN Rotterdam, The Netherlands; c.sloots@erasmusmc.nl; 6Department for Health Evidence, Radboud University Medical Center, 6525 GA Nijmegen, The Netherlands; Iris.vanRooij@radboudumc.nl; 7Faculty of Health, School of Medicine, Witten/Herdecke University, 58448 Witten, Germany; jenetzky@cure-net.de; 8Department of Child and Adolescent Psychiatry and Psychotherapy, University Medical Center of the Johannes-Gutenberg-University, 55131 Mainz, Germany; 9Pediatric Surgery Unit, Cà Foncello Hospital, 31100 Treviso, Italy

**Keywords:** anorectal malformation, anterior anus, anal position index, perineal fistula, anal stenosis, ARM-Net Consortium

## Abstract

In the past, an anteriorly located anus was often misdiagnosed and treated as an anorectal malformation (ARM) with a perineal fistula (PF). The paper aims to define the criteria for a normal anus, an anterior anus (AA) as an anatomic variant, and milder types of ARM such as congenital anal stenosis (CAS) and PF. An extensive literature search was performed by a working group of the ARM-Net Consortium concerning the subject “Normal Anus, AA, and mild ARM”. A consensus on definitions, clinical characteristics, diagnostic management, and treatment modalities was established, and a diagnostic algorithm was proposed. The algorithm enables pediatricians, midwives, gynecologists, and surgeons to make a timely correct diagnosis of any abnormally looking anus and initiate further management if needed. Thus, the routine physical inspection of a newborn should include the inspection of the anus and define its position, relation to the external sphincter, and caliber. A correct diagnosis and use of the presented terminology will avoid misclassifications and allow the initiation of correct management. This will provide a reliable comparison of different therapeutic management and outcomes of these patient cohorts in the future.

## 1. Introduction

Clinicians can be confronted with an abnormal aspect of the anus in newborns. The questions “What is defining a normal anus?” and “When is an anomaly considered an anorectal malformation (ARM)?” remain controversial in the literature. There are a great variability of terms, and the difference between normal anatomy, an anatomic variant, and a pathologic one is often unclear.

These conditions represent the less complex end of the spectrum of ARM but also mild forms like perineal fistulas (PF) can cause significant sequelae, such as chronic constipation [1,2], unnecessary colostomy [1,2,3], overflow-fecal incontinence with severe psychosocial stress [4,5,6], urinary tract infections [7], obstetric injuries [8], and even acute bowel perforation [1,3,9,10] with lethal outcome in the newborn period [1,9,10,11]. It is known that mild forms of ARM may be diagnosed late or even remain undetected [1,12,13]. This is true especially in countries with lower socio-economic facilities [14,15], but even in renowned European centers, between 8.7% and 46% of newborns with ARM are discharged from the birth unit without the correct diagnosis [3,16,17,18]. The rarity of ARMs creates challenges in timely clinical diagnosis. Enhancing clinicians’ awareness and providing clinical tools to differentiate ARM from non-pathological variants would decrease morbidity and potential mortality [10,16,18,19].

Clear criteria are required to determine whether the appearance is still within the normal spectrum, or if a specific ARM is present [1,2,3,9,10,11,16,18,19,20,21,22].

The aim of this paper is to define the criteria for a normal anus, an anterior anus (AA) as an anatomic variant, a mild ARM such as congenital anal stenosis (CAS), and PF. In addition, the ARM-Net Consortium seeks to clarify these entities and outline the diagnostic work-up based on published evidence and personal experiences to provide a timely management approach and avoid complications and unnecessary surgery.

## 2. Materials and Methods

The ARM-Net Consortium, founded in 2010 by European pediatric surgeons, epidemiologists, geneticists, psychologists, and representatives of patient organizations to collect and exchange data and knowledge about ARM to improve clinical care and quality of life of ARM patients by promoting research on genetic, epidemiologic and clinical subjects [23], assembled consecutive working groups on the subject “Normal Anus, AA, and mild ARM” to clarify the differentiation among these entities. After performing an extensive literature search, the working group established a consensus on definitions, diagnostic management, and treatment modalities, providing recommendations in a diagnostic algorithm, which was presented at the annual meeting of the ARM-Net Consortium in October 2021. Finally, a consensus was obtained from the whole ARM-Net Consortium for the present paper in its final form.

## 3. Results

### 3.1. Terminology and Definitions

The ARM-Net consortium agreed on the following terminology and related definitions:Normal anus: Lies in a normal position along the perineum between the fourchette (girls) or scrotum (boys) and the coccyx. It is of normal caliber and circumferentially surrounded by the sphincter muscle complex (Figure 1a).Anterior anus (AA): Considered a normal anatomic variant and defines an anus that is anteriorly located in the perineum, yet fully surrounded by the sphincter muscle complex [24,25,26,27,28,29], and has a normal caliber [30] (Figure 1b). There is no concomitant ARM (such as rectovaginal H-type fistula, Currarino syndrome, etc.).Perineal fistula (PF): Anus is anteriorly located in the perineum and is not completely surrounded by the sphincter muscle complex [29]. It can have a normal or diminished caliber, further defined as non-stenotic or stenotic PF, respectively (Figure 1c).Congenital anal stenosis (CAS): Anus lies in a normal position, completely surrounded by the sphincter muscle complex, but is too narrow. It may be partly covered by a median bar or membrane, usually located at the dentate line [13].


#### 3.1.1. Diagnostic Algorithm

To differentiate among these entities, a diagnostic algorithm (Figure 2) is proposed based on the following questions:

##### How to Determine Whether the Position of the Anus Is Normal?

The normal position of an anus was formerly thought to lie in the midway between the vaginal fourchette and coccyx in girls and the scrotal crease and coccyx in boys [31,32]. Closer observation, using serial measurements, stated that in girls, the anus lies closer to the posterior labial commissure than the midpoint of the perineum [24,26,27,33,34,35,36,37,38,39].

In 1984, the Anal Position Index (API) was proposed by Reisner and colleagues to determine the position of the anal opening in the pelvic floor [38]. API is the ratio of the perineal length divided by the length of the complete posterior pelvic floor, that is, the distance of the fourchette/the scrotal-perineal junction to the center of the anus divided by the distance from the fourchette/scrotal-perineal junction to the tip of the coccyx in girls/boys, respectively (Figure 3).

The mean ratio in neonates was 0.44 ± 0.05 and 0.58 ± 0.06 in females and males, respectively, with significantly lower values in females compared to males [38]. Since the introduction of the API, many authors have proven the API is age and ethnicity-independent [24,26,27,28,33,34,35,36,37,39,40].

An AA was defined when the API results in two standard deviations (SD) below the calculated mean [38]. Using API as a diagnostic tool, an AA was diagnosed with an incidence of 24.6% in otherwise healthy boys and 43.4% of girls, respectively, wherefore it was stated as a common anal abnormality [24]. Conversely, Núñez-Ramos and colleagues investigated the API in more than 1000 newborns in two European hospitals. They reported a significantly lower incidence of AA (2.27–2.84% in females and 1.14–2.10% in males) [27], which corresponds to the expected statistical incidence of 2.28%, assuming a normal distribution. Based on a slight variation of mean and SD, different thresholds for diagnosis of AA can be extracted from data presented in the literature (Table 1) [24,27,33,34,35,36,37,38,39,41,42,43,44,45,46].

Akbiyik and Kutlu evaluated the external genital proportions in 205 pre-pubertal girls and proposed an equation to estimate the expected perineal length, that is 10.314 mm + (0.230 x kg body weight) [47]. The results of this equation are coherent with studies from adult females, where the mean perineal length was described as 25.6 ± 7.3 mm [48] and 21.3 ± 8.5 mm [49].

Besides an anterior position, a congenital lateral or posterior position of the anal opening has never been reported so far.

##### How to Determine if the Anal Opening Is Completely Surrounded by the Sphincter Muscle Complex?

Once the anterior position of the anus is confirmed by API, it is necessary to determine the correlation with the external sphincter in order to differentiate AA from a PF. As previously defined, a normal anus, AA, and CAS are completely surrounded by radiate cutaneous wrinkles that, instead, are lacking ventrally in a PF [24,30] (Figure 1c). To diagnose AA or CAS, the anus must lie in the center of the external sphincter. The ano-cutaneous reflex is evoked in lithotomy position with thighs flexed over the abdomen and the examiner checking whether the anus is fully surrounded by the external contracting sphincter [24,26,27,28,30]. This investigation can usually be done without sedation, by simple stimulation of the skin with a cotton swab. In inconclusive cases, electrostimulation in sedation or general anesthesia, avoiding muscle relaxation [50], is recommended to distinguish between these entities [25,29].

##### How to Determine the Caliber of the Anus in a Newborn?

The calibration of the anus is important to determine the diameter of a PF and rule out CAS. PF has, in most cases, a pathologically small anal caliber. CAS has normal API, complete sphincter encircling, and a diameter smaller than the expected caliber related to the patient’s body weight [21]. CAS is a much rarer form of ARM than PF but presents with similar clinical signs and functional prognosis [51,52].

The examiner’s little finger has been proposed to be the best probe to evaluate the elasticity of an anus in a newborn [21]. However, the little finger of an adult examiner is often too big; therefore, Hegar dilators, whose diameter is expressed in millimeters, should be used [21]. The anal caliber is defined by the Hegar that comfortably fits the anus without resistance and slightly whitens the skin of the anal verge without causing discomfort or pain in an awake neonate. Núñez-Ramos and colleagues reported a size between Hegar 8 and 10 in neonates, with a slight difference between females and males [26,27]. Irrespective of gender, it has been demonstrated that the anal caliber correlates with the body weight in newborns [21,26,27,28]. Thus, an equation has been presented to calculate the expected caliber of a normal anus related to the newborn’s body weight [21].

Equation of the expected anal caliber related to body weight [21]:*Caliber of the anus [*mm*] = 1.34 × body weight [*kg*] + 6.8*

#### 3.1.2. Further Diagnostic Modalities

Further diagnostic investigations to differentiate AA from PF can be considered if the diagnostic methods described above remain inconclusive. Transperineal ultrasound is a non-invasive diagnostic tool that allows evaluation of the anal position with clear visualization of the sphincter muscle complex [53,54,55]. Anorectal manometry has also been proposed to investigate the high-pressure zone of the lower anal canal and to evaluate ventral muscle deficiency [56,57]. Anal endosonography has been used to depict the anatomical integrity of the anal sphincters in children after surgical correction of ARM [58] but has not been used in newborns yet. External phased magnetic resonance imaging (MRI) may provide good information on the anal sphincters, rectum, and pelvic floor musculature [59,60] and has already been used in infants [61,62]. Although MRI may be performed in a feed and wrap technique when the baby is young, it usually requires general anesthesia in infants older than 3 months [63]. The transanal MRI has also been applied in children to assess the anal sphincters after surgery, but it has not yet been used as a preoperative diagnostic tool [64]. Finally, defecography may also be used to evaluate the anorectal angle (see paragraph below) in patients with chronic constipation [65], though it exposes a very young child to a considerable amount of radiation.

### 3.2. Clinical Aspects

What are the possible symptoms of AA, PF, and CAS?

#### 3.2.1. Constipation

In literature, various definitions of constipation are used, and patients with AA, PF, or CAS are often pooled together, leading to unclear information on symptoms and outcomes. Indeed, the former literature on AA reported that this entity is commonly associated with constipation, wherefore surgical repair was often performed [31,66,67,68,69,70,71,72]. However, based on the definition of AA given here, a considerable part of these cases would be nowadays termed as “non-stenotic PF”, as their anus is not completely surrounded by the sphincter. In fact, no significant association of AA with constipation has been observed in several recent reports [24,33,36,37,73]. On the other hand, some authors reported a prevalence of chronic constipation as high as 47% in female and 35% in male AA patients aged 3 months to 12 years [26,27,28]. These patients were conservatively managed, even though the authors reported up to 31% of severe constipation in the very first month of life [26,27,28]. Another study on infants diagnosed with AA showed that constipation rose from 10% in the second month of life to 71.4% at 6 months of age [74].

There is a well-defined, conservatively treated cohort of patients from Finland described as “AA”, which, according to our definitions, also included patients with a non-stenotic PF (“mostly surrounded by sphincter“). When their anal caliber was less than Hegar 12, they were described as AA with concomitant mild AS and treated by anal dilation for 6–8 weeks until Hegar 14 was achieved [75,76]. On follow-up, these patients, older than 7 years, presented a rectoanal inhibitory reflex, anal resting, and squeeze pressures comparable to controls [77]. According to a long-term study, constipation affected 36% of patients versus 13% of a control population (*p* = 0.002); of note, this percentage tended to decline with age [75]. Fecal incontinence is not reported, neither in patients with AA nor in those with non-stenotic PF [75].

#### 3.2.2. Urological and Gynecological Concerns

Because of the greater proximity of the anus to the urethra, a higher rate of urinary tract infections (UTI) in females with AA or non-stenotic PF may be anticipated. Nevertheless, several studies found an equal incidence of lower UTIs compared to controls [73,75]. The gynecologic assessment in post-menarchal girls with AA demonstrated a mean vaginal length of 52 mm (SD: 0.24), comparable with normal reference values for age [78], normal perineal tropism, and normal perineal and vaginal flora [73]. As the study by Duci and colleagues also included patients with non-stenotic PF in their so-called “AA”-population [73], these outcomes favor non-operative management of patients with AA or non-stenotic PF.

In case of pregnancy in these conservatively treated females, the mode of delivery should be individually discussed with the patients in advance. Although vaginal delivery has been reported in women with a history of ARM, such as rectovestibular or rectoperineal fistula, it has been recommended to evaluate the adequacy of the perineal body case by case [79]. By others, cesarean section (CS) has been suggested as the best delivery mode for all patients with AA/PF to avoid perineal tears and obstetric anal sphincter injury (OASI) [8,75]. Ness has recently reported that women in their first labor are at risk for OASI if the perineal length is less than 30 mm [80]. Another study also stated that women with a low API (<0.42) or a short perineum (<40 mm) were prone to traumatic vaginal delivery in primigravidae, with higher rates of episiotomy and instrumented delivery as well as perineal tears [44]. Eventually, OASI can cause burdensome short- and long-term morbidity, affecting the women’s quality of life due to pain, fecal and/or urinary incontinence, and sexual dysfunction. Therefore, CS in patients with either AA or ARM is most likely highly recommended, although each pregnant woman should be individually counseled and the risk factors and benefits of each mode of delivery outlined.

### 3.3. Management

#### Treatment Recommendations for Constipation in Patients with AA, Non-Stenotic PF, or CAS?

In former times, patients with AA were sometimes treated surgically for constipation, but at present, primary conservative management is advocated. Conservative management approaches include stool softeners, laxatives, and transanal bowel management with suppositories or enemas [28]. Based on the long-term outcome of the presented literature, patients with a non-stenotic PF may also be primarily treated by the same conservative management approaches as mentioned for patients with AA [81]. Simple dilations have also been reported as successful in patients with mild stenosis of a PF [13]. Nevertheless, patients with AA or a non-stenotic PF are at risk of developing a rectal cul-de-sac and therefore need to be closely followed to avoid complications later in life.

The mechanism for developing a rectal cul-de-sac was attributed to the incomplete straightening of the rectoanal angle at defecation in case of a ventral malposition of the bowel opening. The descending feces push against the dorsal wall of the anorectal junction, which bulges out, eventually resulting in a dorsal bag or “cul-de-sac”, which might make complete defecation nearly impossible [28,56,67]. Thus, in case of refractory constipation to medical treatment, these patients may eventually benefit from surgical correction [67,70]. Surgical techniques aim to eliminate the cul-de-sac and align the anal canal with the rectum [56,62]. To address that, various techniques such as posterior anoplasty [69,70,71], posterior anoplasty with the complete division of the external sphincter fibers [31], anal transposition [56,57], cutback procedures [66,81,82,83], as well as PSARP [62,84] have been reported in the literature. If surgery is indicated, surgical techniques should aim to preserve the native mechanism of continence, as advocated for any mild form of ARMs [13].

Congenital anal stenosis (CAS) in an orthotopic anus is usually treated by serial anal dilatations, but surgery is required when a median bar, complete membrane, or severe forms are present [13,28]. The rare form of a funnel anus, commonly associated with Currarino syndrome, is characterized by a skin-lined deep anal funnel and the stenotic anal skin-rectum junction, which may also be treated by serial anal dilations [13,85].

In the case of Hegar-dilations, it is paramount that painful dilations are avoided, as they can cause dysfunctional defecation, constipation, and overflow incontinence later in life [86,87].

### 3.4. Genetical Concerns

#### Is Genetic Analysis Warranted in Patients with AA, PF, or CAS?

Patients with a mild ARM (or any other congenital disability) are prone to display additional congenital anomalies. Therefore, every ARM patient should undergo a thorough clinical examination and so-called VACTERL-screening (search for vertebral, cardiac, trachea-esophageal, renal, and limb malformations). Apart from the diagnostic work-up, it is also important to investigate the patient’s family history. Heritability has been reported for patients with either a vestibular or perineal fistula [88]. Concerning patients with AA, Duci and colleagues reported familial occurrence in 5/50 patients of their “AA” population [73]. It has also been reported that risk factors for AA are female gender, high maternal age, and later birth order [74].

Patients with “AA” have also been described as suffering from other congenital malformations or syndromes. However, under the present definitions, it may be these patients were not actually born with a non-stenotic PF. Also, the association of “AA” and perineal groove has been reported [13]. Indeed, Figure 4 shows a patient with “AA”, Pierre-Robin syndrome, and perineal groove.

“AA” was also detected in patients with esophageal atresia, cardiac anomalies, Di-George Syndrome, and Down syndrome [73]. Moreover, a female patient with “AA” was reported to present a disorder of sex differentiation with an accessory phallus, an accessory phallic urethra, and a perineal lipoma [89]. Female patients diagnosed with “ectopic anus”, Hirschsprung disease, and Currarino syndrome at the age of 25 years [90] or affected by X-linked Opitz G/BBB syndrome [91] or Baller-Gerold syndrome [92,93] were also described. An “anterior ectopic anus” was also seen with a rectourethral fistula and a fusiform megalourethra as features of the abdominal muscle deficiency syndrome [94], and in another patient with complete duplication of the bladder, urethra, uterus, and vagina [95]. It was also documented in a female patient with partial trisomy 11q syndrome and deletion 1q44 syndrome [96], in another female with partial monosomy 9p and partial trisomy 18q and stenotic anal opening [97], and in combination with polythelia [98]. The occurrence of propionic acidemia in three siblings with “ectopic anus” in one and a PF in a second sibling from consanguineous parents was suggested for an autosomal recessive genetic inheritance [99]. “AA” was also documented as a feature of an unknown mandibulofacial dysostosis syndrome occurring with duodenal and biliary atresia associated with facial, thyroid, and auditory apparatus abnormalities [100]. Furthermore, “AA” was observed in patients with Juberg-Hayward syndrome [101]. In combination with the cardinal manifestations of aplasia cutis and epibulbar dermoid, “AA” was mentioned with other features such as laryngomalacia, microcephaly, and significant developmental delay to describe an oculo-ectodermal syndrome with possibly recessive inheritance [102].

These examples highlight the importance of investigating the family history and performing a physical examination and VACTERL-screening of patients with mild ARM, but also with mere AA, to not miss possible syndromes or other congenital anomalies. However, because of the noxious side effects of radiation, especially in children, radiologic examinations should be only performed upon abnormal clinical or ultrasound/MRI findings. The concept that AA might display an increased risk for associated malformations or if this current assumption is based on biased data from the past when AA and non-stenotic PF-patients were not differentiated from each other requires further studies.

## 4. Discussion

The term “ectopic anus” was first used in 1958 to describe an anal opening anterior to the external sphincter [66]. A full twenty years later, Hendren reported on patients with an “anterior anal opening” treated with anoplasty for persistent constipation [69].

According to the present knowledge, an anteriorly located anus (AA) is defined as a normal variant of an anus located more anteriorly along the perineal body with a normal caliber, completely surrounded by the anal sphincter complex [24,25,26,27,28,29]. Yet, the occurrence of AA in patients with a genetically recognized syndrome, as well as the observed rate of stenosis of AA [13,28,75], may suggest that AA should be considered a mild form of ARM. The terms “anterior anus”, “anterior displacement of the anus”, ”anteriorly displaced anus”, “anteposition of the anus”, “anterior ectopic anus”, ”ectopic anus”, “anterior perineal anus”, or “anus perinei ventralis” have been variably used to describe either patients with AA according to the present definition, or patients with a bowel opening not completely surrounded by the sphincter muscle complex, which we propose to term consistently as PF [13,24,25,26,27,28,29,30,31,33,34,35,36,37,38,39,41,44,45,46,53,55,56,57,61,62,66,67,68,69,70,71,72,73,74,75,77,82,83,84,89,90,91,92,93,94,95,96,97,98,99,100,101,102,103,104,105,106,107,108,109,110]. Because of this unclear terminology, management and outcome parameters could not be reliably compared.

An anterior position of an anus is diagnosed when the API is 2 SD below the calculated mean [38,40]. However, the calculations of mean and SD are based on the concept of a normal distribution, but in physiological conditions, a skewed distribution more often applies. Percentile values would be more helpful, and thus, future studies on patients with AA or API measurements should also report non-parametric values of the API besides mean and SD, to reduce this bias. The reliability of the API measurements performed by different investigators has not been evaluated yet. By comparing the measurements of different investigators in our clinics, we identified this as a possible point of concern. Therefore, it will be the objective of a future ARM-Net study.

We consider AA a normal anatomic variant of the anus with a normal anal canal. Recent long-term outcomes proving normal bowel control, normal frequency of lower urinary tract symptoms, as well as normal perineal tropism and perineal and vaginal colonization, all favor non-operative management of the affected children. In this context, we fully respect the groundbreaking work from the Helsinki group concerning the conservative treatment of patients with mild forms of ARM, even though patients with the anus not completely surrounded by the external sphincter muscle were included in their “AA” group [75]. If uniform definitions are used, it will be possible to reliably compare the series of patients from different centers in the future.

Procedures, such as anal dilations, are not required for AA unless stenosis is encountered, but might be of benefit in the case of CAS [13,75]. Painful dilations must be avoided, as they can cause dysfunctional defecation, constipation, and overflow incontinence later in life [86,87].

According to present knowledge, the “fistula” in ARM represents an ectopic anal canal and should be preserved as far as possible to improve the chance for fecal continence [13,77,81,111].

AA may be a hint to investigate for suspected syndromes. In families with ARM-affected members, AA may signify a possible genetic involvement. It is unknown whether AA, according to the present definition, may be associated with other congenital anomalies. Therefore, a thorough physical examination and VACTERL-screening are suggested in every child with AA as in any other ARM patient. Future studies will reveal whether children with AA may present a greater risk for other congenital anomalies or if the management can be limited to an accurate physical examination.

Environmental factors also play a role in the etiology of AA. Prenatal phthalate metabolite exposure was documented to reduce the anogenital distance in newborns with unknown consequences [112,113]. Exposure to the pesticide dichlorodiphenyltrichloroethane (DDT) metabolite 1,1-dichloro-2,2-bis (p-chlorophenyl) ethylene (DDE) during the first trimester of pregnancy has already been associated with a significant reduction of the API in boys (ß = −0.02; *p* = 0.02) [45]. However, a more recent study could not confirm an association between maternal DDT and DDE isomers and the anogenital distance between boys and girls at birth but noted a significant association between the maternal serum concentration of o,p’-DDE, the breakdown product of 1,1,1-trichloro-2-(2-chlorophenyl)-2-(4-chlorophenyl) ethane (o,p’-DDT) at delivery with a shorter ano-fourchette distance at 1-year of age [42]. These environmental factors might cause a greater variability of API measurements than expected [114,115].

Children with AA have been reported to experience normal bowel control and a normal stooling pattern. In some cases, however, defecation disorders may develop in certain patients. Indeed, AA was often seen as the cause of chronic constipation in many patients who presented with a large cul-de-sac and were candidates for surgical management. However, cases of AA and PF are not clearly distinguished from each other in most of the papers, leading to pooled therapeutic results. Conservative management such as diet, stool softeners, laxatives, and rectal irrigation should always be the first-line treatment option in a child with AA who develops signs of constipation. As the onset of constipation has been reported in patients with AA already during infancy, we advise closely following children with AA in their first year of life. According to a long-term study by Kyrklund and colleagues, constipation declines with age and can be successfully managed by medical treatment [75]. Both Kyrklund and Duci, who included non-stenotic PF-patients with an anal opening mostly surrounded by sphincter in their AA population, proved that these patients have a similar satisfactory long-term outcome regarding bowel function under conservative management [73,75].

Hence, we suggest this patient group should be conservatively managed, except (i) in case of a stenotic PF (we recommend Hegar ≤ 8 in a term newborn), (ii) in case of a PF surrounded by the sphincter less than 75% of the anal circumference, or (iii) in patients with AA or non-stenotic PF who develop refractory constipation on conservative management.

Until now, no studies compare obstetric complications in mothers with conservatively versus surgically treated PF, outlining the need for individual counseling concerning the mode of delivery. Future studies addressing the mode of delivery are mandatory. They might change the recommendations given here for CS in women with a history of ARM and the conservative treatment of female patients with non-stenotic PF or AA.

## 5. Conclusions

We seek to clarify the differentiation between anatomic variants and mild forms of ARM (PF, CAS) by providing a clinical decision tool to determine which condition is present in an abnormally looking anus at birth. Part of the presented algorithm is the proposal of a binding terminology for AA and PF. This common language is a prerequisite for further research comparing the results of different treatments, namely conservative and surgical, and for better defining the best way to manage the affected children.

## Figures and Tables

**Figure 1 children-09-00831-f001:**
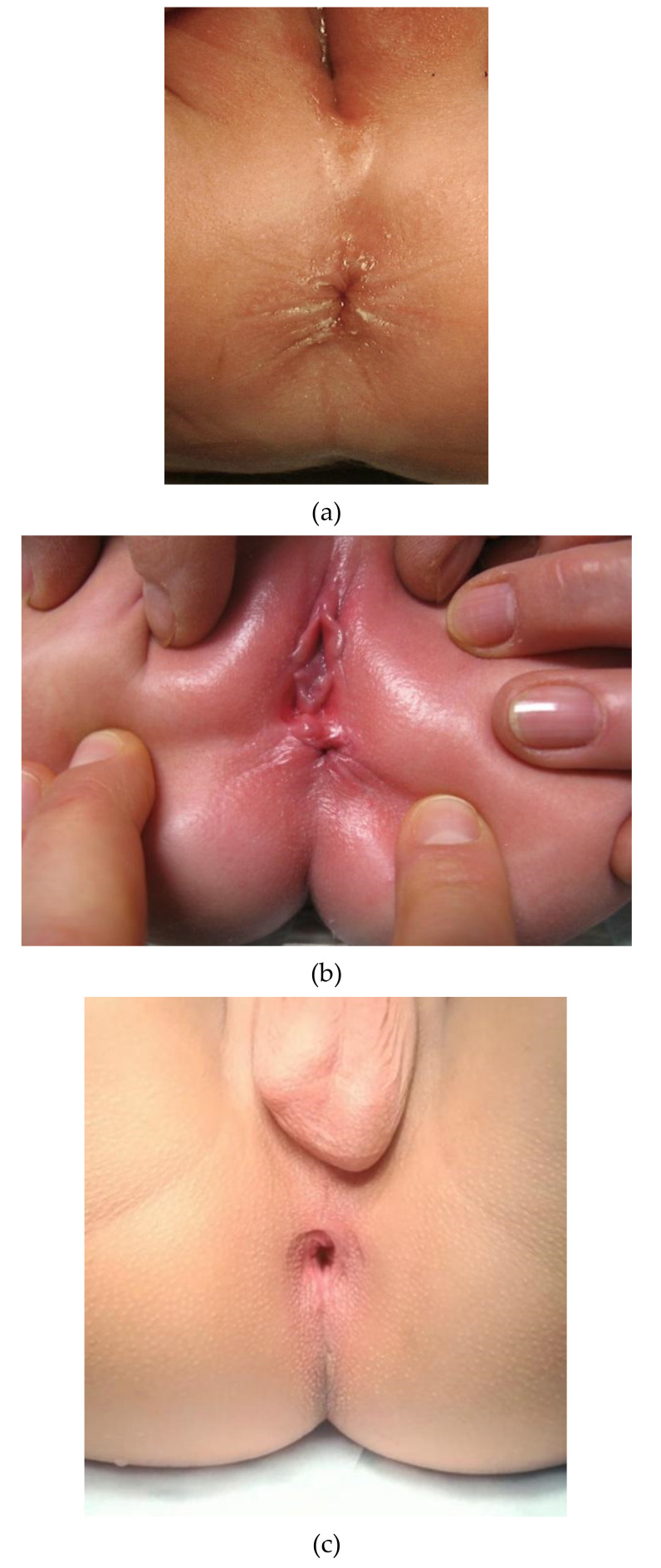
(**a**) Picture of a normal anus in a female infant. (**b**) Picture of an anterior anus in a female infant. Note the proximity of the anal opening to the external genitalia. (**c**). Picture of a non-stenotic perineal fistula. The sphincter muscle complex does not encircle the anal opening along the anterior margin.

**Figure 2 children-09-00831-f002:**
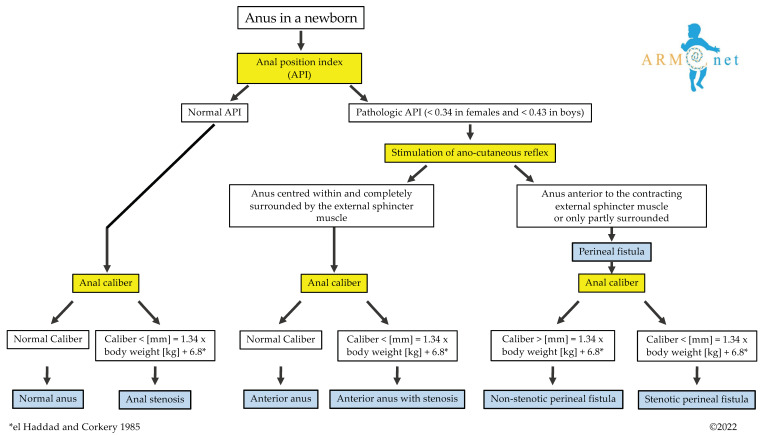
The algorithm shows the diagnostic pathway that leads to diagnosing a normal anus, anterior anus with or without stenosis, congenital anal stenosis, and perineal fistula (stenotic or non-stenotic). * The anal caliber is measured according to the equation presented by el Haddad and Corkery [21].

**Figure 3 children-09-00831-f003:**
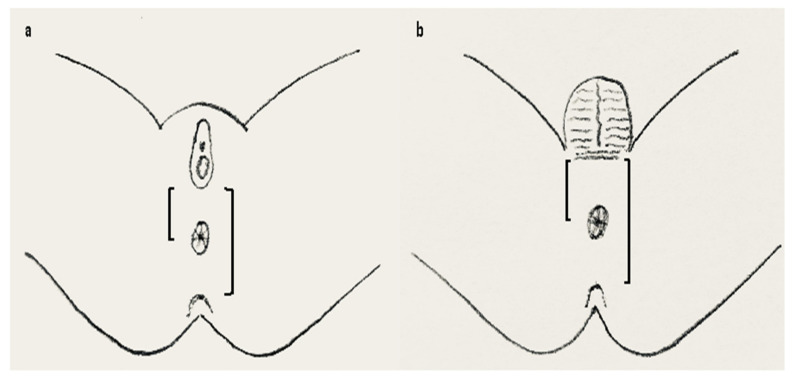
API measurement in females (**a**) and males (**b**). API is calculated by dividing the distance between the fourchette (**a**) or scroto-perineal fold (**b**) to the center of the anus with the distance between fourchette (**a**) or scroto-perineal fold (**b**) to the tip of the coccyx marked on a transparent tape or flexible ruler. Anterior displacement is diagnosed when the API is <0.34 in females and <0.43 in boys (overall mean minus 2 standard deviations as presented by Sharma et al. [40]). API = anal position index.

**Figure 4 children-09-00831-f004:**
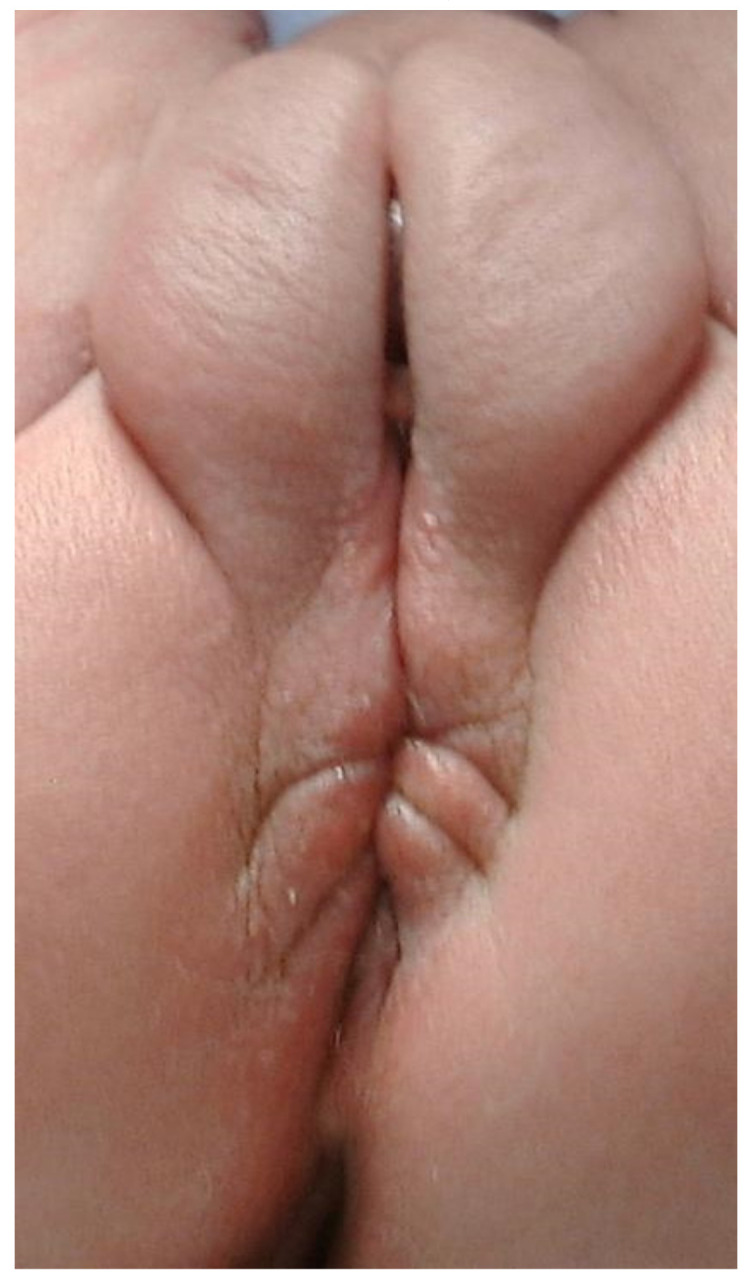
Picture of a female newborn with AA and a perineal groove. The patient also has Pierre-Robin syndrome.

**Table 1 children-09-00831-t001:** Results of the API-values and the criteria for anterior displacement according to published studies (adapted from Sharma et al. [40]). Additional API values presented in the literature have been included.

**Study**	**Age Group**	**Country**	**N**	Mean Values of API		Anterior Displacement -2 SD	
				Male (SD) (*n*)	Female (SD) (*n*)	Male	Female
1. Reisner et al., 1984 [38]	Newborn	Israel	200	0.58 (0.06) (100)	0.44 (0.05) (100)	<0.46	<0.34
	4-18 months		30	0.56 (0.4) (15)	0.40 (0.06) (15)		<0.28
2. Bar-Maor and Eitan 1987 [33]	3 days–12 years	Israel	104	0.56 (0.10 *) (74)	0.39 (0.09) (30)	<0.36	<0.21
	Constipated 3d-12y		34	0.58 (0.09) (23)	0.4 (0.07) (11)	<0.4	<0.26
3. Genç et al., 2002 [36]	Newborn	Turkey	60	0.53 (0.05) (26)	0.46 (0.08) (34)	<0.43	<0.30
4. Mohta and Goel 2004 [37]	Newborn–3 years	India	387	0.43 (0.05) (300)	0.37 (0.06) (87)	<0.33	<0.25
5. Herek and Polat 2004 [24]	Newborn–10 years	Turkey	357	0.51 (0.08) (191)	0.36 (0.08) (166)	<0.35	<0.20
6. Rerksuppaphol et al., 2008 [39]	Newborn	Thailand	403	0.51 (0.07) (203)	0.38 (0.08) (200)	<0.37	<0.22
7. Davari and Hosseinpour 2006 [35]	Newborn	Iran	400	0.54 (0.07) (200)	0.45 ** (0.08) (200)	<0.40	<0.29
8. Patel et al., 2018 [43]	Newborn and infants	India	65	0.53 (0.07) (31)	0.36 (0.07) (34)	<0.39	<0.22
9. Chan et al., 2009 [34]	Neonates	Taiwan	200	0.54 (0.03) (100)	0.40 (0.04) (100)	<0.48	<0.32
	5–19 months		30	0.53 (0.02) (15)	0.39 (0.06) (15)	<0.49	<0.27
10. Núñez-Ramos et al., 2011 [27]	Newborn	Spain	529	0.53 (0.06) (262)	0.40 (0.05) (267)	<0.41	<0.30
	Older constipated patients		64	0.47 (0.10) (26)	0.36 (0.10) (38)	<0.27	<0.16
11. Núñez-Ramos et al., 2011 [27]	Newborn	Italy	483	0.51 (0.06) (237)	0.39 (0.08) (246)	<0.39	<0.23
12. Torres-Sanchez et al., 2008 [45]	Newborn–Infants	Mexico	71	0.6 (0.07) (37)	0.5 (0.07) (34)	<0.46	<0.36
Sharma et al., 2021 [40]	overall mean of all studies:			0.51 (0.04)	0.40 (0.03)	<0.43	<0.34
13. Rizk and Thomas 2000 [44]	Adult women	UAE	212		0.49 (0.12) (212)		<0.25
14. Bornman et al., 2016 [42]	Newborn	South Africa	659	0.58 (0.59) (336) ***	0.46 (0.61) (323) ***	<0.46	<0.33
15. Alemrajabi et al., 2019 [41]	Adults	Iran	63	0.53 (0.11) (48)	0.45 (0.16) (15)	<0.31	<0.13
16. Tufekci and Yesildag 2021 [46]	Neonates	Turkey	405	0.52 (0.05) (230)	0.39 (0.4) (175)	<0.42 ****	<0.31 ****

The values defining anterior displacement in italic print were calculated as minus 2 SD from the mean. Sharma et al. evaluated the overall mean API from data retrieved from the studies 1-12 [40]. * Bar-Maor and Eitan [33] indicated a SD of 0.10 in Figure 2 in their original publication, while the text there reads SD 0.20, which was quoted by Sharma et al. [40]. **According to the original publication from Davari and Hosseinpour [35] the mean API in females is 0.45 and not 0.42 as quoted by Sharma et al. [40]. *** Bornman et al. [42] showed an API of 58.3 (SD 5.9) for males and 46.8 (SD 6.1) for females in their Table 1 which is considered a calculation error. **** Tufekci and Yesildag [46] originally diagnosed an AA in their study when the API was below the 5th percentile, which was 0.33 in girls and 0.43 in boys, respectively. AA = anterior anus; UAE = United Arab Emirates. A recent meta-analysis evaluating the utility of API outlined an overall mean API of 0.40 ± 0.03 in females and 0.51 ± 0.04 in males, with abnormal values less than 0.30–0.34 in females and 0.41–0.46 in males (Table 1) [40].

## Data Availability

Not applicable.

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
