# Peer review of "Clinical Differentiation between a Normal Anus, Anterior Anus, Congenital Anal Stenosis, and Perineal Fistula: Definitions and Consequences—The ARM-Net Consortium Consensus"

_children, 2022, doi:10.3390/children9060831_

Round 1
Reviewer 1 Report
This is a well written narrative review on a delicate and narrow topic.
The algorithm image (Figure 2) is incomplete/broken - this should be remedied.
Figure 3 - is this adapted or a new creation ?
Table 2 - adapted material but use the same font as in the rest of the article.
Author Response
We deeply thank the referee for his/her comments and appreciate his/her support to improve the manuscript.
The manuscript was changed, according to the comments.
We apologize for the bad quality of “Figure 2” in the original submitted manuscript. “Figure 2” have been replaced by a new version with the correct font type of the manuscript. Accordingly, also “Table 1” was changed into the font type of the manuscript. We uploaded figure 2 separately. Please, see the attachment.
Concerning the question about “Figure 3” (adapted or created de novo) we would like to clarify that this figure was newly created by us to explain the API measurement.

Reviewer 2 Report
The manuscript "Clinical Differentiation between a Normal Anus, Anterior Anus, Congenital Anal Stenosis, and Perineal Fistula: Definitions and Consequences - the ARM-Net Consortium Consensus" is an interesting manuscript with the aim of summarize the criteria for differentiation between a normal anus, an anterior anus (AA) and milder types of ARM such as congenital anal stenosis (CAS) and PF. The manuscript relies on the ARM-Net Consortium and clearly defines a subset of characteristics that can identify and diferentiate between the clinical conditions specified.
The manuscript is well written and summarizes the most relevant aspects.
Author Response
We thank the reviewer for his/her comment and endorsement to our paper
Please see the attachment
